# Inhibition of Multifunctional Protein p32/C1QBP Promotes Cytostatic Effects in Colon Cancer Cells by Altering Mitogenic Signaling Pathways and Promoting Mitochondrial Damage

**DOI:** 10.3390/ijms25052712

**Published:** 2024-02-27

**Authors:** Carlos Alejandro Egusquiza-Alvarez, Angela Patricia Moreno-Londoño, Eduardo Alvarado-Ortiz, María del Pilar Ramos-Godínez, Miguel Angel Sarabia-Sánchez, María Cristina Castañeda-Patlán, Martha Robles-Flores

**Affiliations:** 1Departamento de Bioquímica, Facultad de Medicina, Universidad Nacional Autónoma de México (UNAM), Mexico City 04510, Mexico; cegusquiza3@gmail.com (C.A.E.-A.); patrick-an@hotmail.com (A.P.M.-L.); eduralv.o@gmail.com (E.A.-O.); mike_sarabia@hotmail.com (M.A.S.-S.); cristi_ccp@bq.unam.mx (M.C.C.-P.); 2Departamento de Microscopía Electrónica, Instituto Nacional de Cancerología, Secretaría de Salud, Mexico City 14080, Mexico; pilyrg@gmail.com

**Keywords:** p32/gC1qR/C1QBP/HABP1, colon cancer, M36 inhibitor, cytostatic effects, mitochondrial dynamics, mitochondrial damage, MAPK signaling pathway, PI3K/Akt/mTOR signaling pathway

## Abstract

The protein p32 (C1QBP) is a multifunctional and multicompartmental homotrimer that is overexpressed in many cancer types, including colon cancer. High expression levels of C1QBP are negatively correlated with the survival of patients. Previously, we demonstrated that C1QBP is an essential promoter of migration, chemoresistance, clonogenic, and tumorigenic capacity in colon cancer cells. However, the mechanisms underlying these functions and the effects of specific C1QBP protein inhibitors remain unexplored. Here, we show that the specific pharmacological inhibition of C1QBP with the small molecule M36 significantly decreased the viability rate, clonogenic capacity, and proliferation rate of different colon cancer cell lines in a dose-dependent manner. The effects of the inhibitor of C1QBP were cytostatic and non-cytotoxic, inducing a decreased activation rate of critical pro-malignant and mitogenic cellular pathways such as Akt-mTOR and MAPK in RKO colon cancer cells. Additionally, treatment with M36 significantly affected the mitochondrial integrity and dynamics of malignant cells, indicating that p32/C1QBP plays an essential role in maintaining mitochondrial homeostasis. Altogether, our results reinforce that C1QBP is an important oncogene target and that M36 may be a promising therapeutic drug for the treatment of colon cancer.

## 1. Introduction

Colorectal cancer constitutes the second most common cause of cancer deaths worldwide [1]. Many therapeutic strategies have been developed to treat this lethal pathology, including surgery, chemotherapy, radiotherapy, and monoclonal antibodies [2]. However, these therapies do not work in many patients, especially in the advanced stages of the disease. Therefore, the development of new strategies that serve as a part of specific targeted therapies for each patient is crucial.

The protein p32 (also known as gC1qR, C1QBP, or HABP1) is a multicompartmental and multifunctional molecular marker found to be overexpressed in several adenocarcinomas, including colorectal cancer [3,4,5]. C1QBP is an acidic donut-shaped homotrimer protein that is highly conserved from yeast to humans, suggesting that this protein has important functional roles for the protein in eukaryote organisms [6]. Although its canonical location is the mitochondria [7,8], this protein is also found in the cytosol, nucleus, and is possibly secreted depending on the cell context [3,9]. Due to its high structural flexibility [10], C1QBP can bind many different and functionally divergent ligands, including hyaluronic acid, protein kinase C (PKC) [11,12], ASF/SF2 [13,14], and p53 [15]. Interactions with C1QBP can activate, inhibit, or have no effects on the binding partner. In this way, the protein can regulate as many functions as the ligands that it binds, functioning as a chaperone, scaffold protein, or regulatory subunit of other proteins [3]. Although it is difficult to assign a specific function to C1QBP, the most salient cellular role of this protein is in regulating the biosynthesis of mitochondrial genome-codified proteins and, consequently, the mitochondrial metabolism (mainly oxidative phosphorylation) [16,17], morphology, and dynamics [18,19]. Another function assigned to this protein is the regulation of apoptosis through the modulation of the permeability transition of mitochondrial (PTM) pore [20,21] and its interaction with other mitochondrial proteins involved in apoptosis, such as Hrk [22] and smARF [23]. The effects exerted by this protein can be pro-apoptotic or anti-apoptotic, depending on the cellular context [3]. Other key functions demonstrated for this protein include the modulation of splicing acting as a regulatory subunit of the splicing factor ASF/SF2 [14] and the regulation of several transcription factors such as TFIIB [24], CBF/NF-Y [25], and p53 [15]. This protein also plays critical roles in immune responses and inflammation by regulating the complement system, promoting the generation of bradykinin, and inhibiting the secretion of the cytokines IL-1 and IL-12 [4,26]. Several studies have also reported the crucial role of this protein in regulating several cell signaling pathways at different levels, including as a ligand, as a receptor, and as a modulator of key cell signaling proteins such as PKC [3].

Interestingly, this multifunctional protein was found to be overexpressed in several types of epithelial malignant tumors and cancer cell lines compared with their non-malignant counterparts [27,28,29,30,31,32]. A plethora of studies on the gain and/or loss of function have shown that p32/C1QBP is involved in promoting several malignancy traits in different cancer types [19,21,28,33,34,35,36]. Hence, the accumulated evidence strongly suggests that C1QBP is an oncogenic protein.

We previously demonstrated that the depletion of p32/C1QBP in colon cancer cells negatively affected their migration capacity, oxidative stress resistance, chemoresistance to CCI-779 and 5FU drugs, and clonogenic and tumorigenic capacity [28]. We observed that the basis of these oncogenic functions lies in the fact that the protein promotes the activation of the Akt-mTOR signaling pathway and regulates the expression of several genes related to malignant phenotypes such as HAS-2 and PDCD4. These data suggest, therefore, that C1QBP could be a therapeutic target for developing new treatment approaches for colon cancer. This study aimed to evaluate the effects of a drug with p32/C1QBP as a specific target to understand the molecular and cellular mechanisms underlying the oncogenic functions of C1QBP in colon cancer cells. To this end, we used the M36 inhibitor developed by Yenugonda et al. through a pharmacophore model of the association of p32/C1QBP with its ligands C1q and Lyp1 [37]. This inhibitor was shown to mimic the effects of knocking down C1QBP in glioma cells and have an antiproliferative effect on this cancer cell model [37]. However, to date, no studies have evaluated the effects of this drug in colon cancer cells so far, and the molecular mechanisms underlying its biological effects remain to be elucidated.

Here, we show that p32/C1QBP induces a pro-growth effect on colon cancer cells by regulating the activation of signaling pathways triggered by growth factors and functioning as a key modulator of mitochondrial integrity and dynamics. Remarkably, our results point to an M36 inhibitor as a possible therapeutic agent for the treatment of colon cancer.

## 2. Results

### 2.1. Pharmacological Inhibition of the p32/C1QBP Protein Negatively Affects the Viability of Colon Cancer Cells

We have previously demonstrated that the p32/C1QBP protein is an essential promoter of migration, chemoresistance, clonogenic, and tumorigenic capacity in colon cancer cells [28]. However, the mechanisms underlying these functions and the effects of specific C1QBP protein inhibitors remain unexplored. In this study, we aimed to evaluate the effects of a drug with p32/C1QBP as a specific target to understand the molecular and cellular mechanisms underlying the oncogenic functions of C1QBP in colon cancer cells. To this end, we used the M36 inhibitor developed by Yenugonda et al. based on a pharmacophore model of the association between p32/C1QBP and its ligands, C1q and Lyp1 [37].

We first evaluated whether M36 had an effect on the viability of RKO, HCT116, SW480, and SW620 cell lines. We found that the M36 inhibitor significantly decreased the viability rate of the four evaluated colon cancer cell lines in a dose-dependent manner (Figure 1A). Interestingly, the most marked effect was observed in the RKO colon cancer cell line (IC50 = 55.86 μM), which presented the highest C1QBP expression level among the evaluated cell types [28]. Additionally, we demonstrated that the effects exerted by M36 on the viability rate were significantly higher than those exerted by the vehicle DMSO at the same concentration (Figure 1B). However, when we examined the effects of the inhibitor on the viability of non-malignant colon cell line 112CoN (with a relatively low expression of C1QBP), we did not observe any significant differences between the control vehicle and the M36 inhibitor effects. Congruent with this result, treating RKO cells featuring the knockdown of p32/C1QBP did not affect cell viability compared with the control vehicle. Therefore, we can conclude that the pharmacological inhibition of C1QBP with M36 induced a decrease in the viability rate of colon cancer cells with high expression levels of p32/C1QBP.

### 2.2. Pharmacological Inhibition of p32 Protein Significantly Decreased the Clonogenic Capacity of Colon Cancer Cells

To further investigate the effects exerted by M36 on the malignancy traits of colon cancer cell lines, we performed a colony formation assay to determine the effects of the inhibitor on the clonogenic ability of these cells. Again, we found that treating the RKO, HCT116, SW480, and SW620 cell lines with the inhibitor M36 (at their respective IC50 concentrations) significantly decreased the clonogenic capacity of the cells compared with that in the group treated with the vehicle at the same concentration (Figure 2). Again, we found the most marked effect on the RKO cell line. Interestingly, not only the number but also the size of the colonies were affected by treatment with M36 in each case, strongly suggesting an anti-proliferative effect induced by M36 on colon cancer cell lines.

### 2.3. Pharmacological Inhibition of p32 Protein Induces a Cytostaic but Not Cytotoxic Effect on RKO Colon Cancer Cells

We next explored whether M36 had a cytostatic or cytotoxic effect on the RKO cell line. As shown in Figure 3A, in the group treated with the M36 inhibitor, we did not find any evidence of apoptotic or necrotic cells such as cellular shrinkage, rounding, or detachment from the substrate among RKO cells incubated for 72 h in the absence or presence of M36 (at IC50 = 56 µM). However, we observed a clear decrease in cell density compared with that observed in the control and vehicle-treated cells. In addition, we found no morphological markers of cell death in the cells treated with M36 compared with the vehicle or non-treated control since M36 did not induce the activation of PARP or Caspase-3 via proteolytic cleavage (Figure 3B). We also did not find evidence of cell death or detached cells on the supernatant. These results, therefore, strongly suggest that M36 provoked a cytostatic but not a cytotoxic effect in treated colon cells. In line with this result, treating RKO cells with M36 significantly increased the level of p21 (Figure 3C), a protein closely related to cell cycle arrest and the inhibition of cell growth [38].

To demonstrate that a decrease in the proliferation rate caused a decrease in the number of cells, we employed a Carboxyfluorescein Diacetate Succinimidyl Ester (CFSE) proliferation assay. Proliferation was measured by labeling the cells with fluorescent dye to track the generations of cells, since the associated fluorescence signal decreased by one-half with each cell division cycle. Figure 3D indicates that M36 treatment for 6 days negatively affected the proliferation of RKO cells compared with the control vehicle and non-treated cells. This effect is visualized in the figure as a retention of the fluorescent compound CFSE in cells treated with M36 (Figure 3D). Altogether, these results reinforce the notion that the pharmacological inhibition of the p32/C1QBP protein with M36 has a cytostatic effect on colon cancer cells.

### 2.4. Pharmacological Inhibition of the p32/C1QBP Protein Negatively Affects the Rate of the Activation of Mitogenic Signaling Pathways

It is well documented that C1QBP promotes the activation of several signaling pathways downstream from receptor tyrosine kinases (RTKs) such as the PI3K-AKT-mTOR and MAP kinases pathways [28,34,36,39]. In agreement with this phenomenon, we previously demonstrated that the knockdown of p32/C1QBP inhibits mTOR activity in both RKO and SW480 colon cancer cells [28]. Hence, we sought to evaluate whether treating RKO colon cancer cells with the M36 inhibitor could also affect the activation rate of the Akt-mTOR and MAPK pathways. To achieve this goal, we stimulated RKO cells with FBS at 10% for 3 and 6 h in the presence of M36 or the same concentration of the vehicle. Then, we evaluated the activation rates of key proteins involved in these pathways. We found that after 6 h of stimulation with FBS at 10%, the group treated with M36 displayed a significantly lower activation rate of AKT protein via phosphorylation than that in the group treated with the vehicle (Figure 4A). Similar results were obtained under activation via the phosphorylation of mTORC1 (Figure 4B), the typical mTOR substrates p70-S6K and 4-EBP (Figure 4C,E), and ERK (Figure 4D) proteins, confirming that the M36 inhibitor can obviate the role of the p32/C1QBP protein in promoting the activation of Akt-mTOR and MAPK pathways.

Indeed, mTOR is an essential regulator of cell metabolism, proliferation, and survival, in addition to regulating autophagy. Since it was reported that autophagy is induced by multiple anticancer agents (specially mTOR inhibitors) as a tumor-survival-promoting mechanism [40,41], we also assessed whether the mTOR inhibition induced by M36 treatment would affect the activation/phosphorylation status of serine/threonine kinase Unc-51- like kinase-1 (ULK1), which is crucial in inducing autophagy/mitophagy. Remarkably, p32/C1QBP was identified as a key regulator of ULK1′s stability [42] and is involved in inducing mitochondrial autophagy to maintain healthy cellular homeostasis [42,43]. We must consider that ULK possesses different phosphorylation sites that regulate its ability to initiate or not initiate the autophagy process or repress it. The sites Ser317, Ser555, and Ser777 are commonly related to inducing the autophagosome formation via AMPK activity [44], while the Phospho-ULK Ser757, induced by mTORC1 or p38 MAPK [45], represses the autophagy process, thereby avoiding the regulation mediated by AMPK. Under our conditions, stimulation solely with 10% FBS induced the mTORC1 activity and, consequently, the phosphorylation of ULK at the Ser757 site (repressing ULK1 activity), as shown in Figure 4F. Interestingly, although M36 inhibited mTORC1, thus decreasing Phospho-p70S6K and Phospho-4EBP levels, there was no effect on Phospho-ULK Ser757 levels (Figure 4F). Consistently, we found no changes in macroautophagic markers such as the LC3 II/I ratio and p62 (Appendix A), indicating that basal levels of autophagy were maintained when p32/C1QBP was inhibited. These results also suggested that the inhibition of C1QBP impaired its cytoprotective role in inducing Ulk1-dependent autophagy/mitophagy.

### 2.5. Pharmacological Inhibition of the p32/C1QBP Protein Negatively Affects the Level of Expression Levels of Proteins Involved in Mitochondrial Dynamics and Induces Mitochondrial Damage

One of the canonical functions of p32/C1QBP in different cell types, including tumoral cells, is the regulation of mitochondrial morphology, integrity, dynamics, and autophagy by removing the dysfunctional mitochondria [18,19,46]. Therefore, we first explored whether the M36 inhibitor of C1QBP affected these mitochondrial functions. To accomplish this goal, we evaluated the effects of the inhibitor at the level of expression for three key proteins in the regulation of mitochondrial dynamics. First, we evaluated the expression levels of DRP1, a protein involved in mitochondrial membrane constriction during the fission process [47]. We found that cells treated with M36 presented decreased levels of DRP1 compared with cells treated or not treated with the vehicle (Figure 5A). We also evaluated the levels of OPA1 and Mitofusin-2, two proteins implicated in mitochondrial membranes fusion [47,48]. Interestingly, we also observed a decrease in the levels of these proteins involved in mitochondrial fusion when RKO cells were treated with M36 compared with the levels when cells were treated with the vehicle or were not treated (Figure 5B,C).

To explore the impacts of p32/C1QBP inhibition on mitophagy, we examined the effects on the PINK–Parkin pathway, which plays an important role in regulating mitophagy [49,50]. It was demonstrated that C1QBP is involved in mitochondrial autophagy in removing the dysfunctional mitochondria by regulating Parkin [51]. In line with this function, Figure 5D shows that C1QBP inhibition significantly decreased Parkin levels while not affecting total PINK total levels (Figure 5E), which would impede the damaged mitochondria from being removed.

The mitochondrial membrane potential and morphology are considered key information parameters for the mitochondrial functional state. These parameters can be studied using fluorescent dyes (“probes”) such as tetramethylrhodamine methyl ester (TMRM) and MitoTracker (MT) dyes [52]. We monitored the mitochondrial integrity through immunofluorescence and microscopy using a MitoTracker Red CMXRos probe as an indicator of mitochondrial membrane potential alterations in the mitochondrial membrane. Interestingly, we found that cells treated with M36 featured a decrease in the median fluorescence intensity (MFI) of MitoTracker per cell compared with the MFI of cells that were treated with the vehicle or were not treated (Figure 5F). We also found that the distribution of the MitoTracker was more diffuse in cells treated with M36 than in control cells that were treated with the vehicle or were not treated. A higher fragmentation rate was also observed in the cells treated with M36 (Figure 5G). To confirm mitochondrial damage, we observed the morphology of the mitochondria with a transmission electron microscope (TEM). The resulting images were recorded using a digital camera, as shown in Figure 6. In the control cells (with or without the vehicle), it can be seen that the number of mitochondria was stable, the structure of the mitochondrial ridge was normal, and the mitochondria did not show evidence of swelling following M36 treatment. In contrast, in the M36-treated cells, the mitochondria’s swelling increased, the mitochondria’s vacuolization was more severe, and the structure of the mitochondrial ridges became disordered or disappeared. All these results strongly suggested that many key functions carried out by mitochondria could be affected in colon cancer cells as a result of p32/C1QBP inhibition with M36.

## 3. Discussion

In the present work, we demonstrated that the pharmacological inhibition of p32/C1QBP, using the small molecular inhibitor M36, induced, in a dose-dependent manner, a significant decrease in the viability, proliferation rate, and clonogenic capacity of different colon cancer cell lines overexpressing C1QBP, consistent with our previous studies highlighting p32/C1QBP as an important oncogene that promotes a malignant phenotype in colon cancer cells [28]. By investigating the underlying mechanisms of C1QBP inhibition using its specific inhibitor M36, we not only confirmed the important role that p32/C1QBP plays in the positive modulation of oncogenic pathways downstream from RTKs but also found that C1QBP inhibition with M36 affects the crucial role of C1QBP in the maintenance of mitochondrial integrity and dynamics in colon cancer cells.

Recent evidence obtained in different laboratories showed the crucial role of p32/C1QBP as a promoter of activating signaling pathways downstream of RTKs [28,34,36,39]. Several authors have demonstrated in various cancer types that C1QBP can positively regulate the autophosphorylation of different RTKs such as EGFR, VEGFR, and IGFR in various cancer types by binding to lipid rafts containing these receptors [36,39]. There is also some evidence showing that CD44 is an essential mediator for this function of C1QBP. However, the precise molecular mechanism and interactions underlying this effect on RTKs’ autophosphorylation are yet to be completely understood. The previous results suggest that the binding of M36 to C1QBP considerably affects the translocation of the protein to the lipid rafts’ domains in the cytoplasmic membrane and/or the interaction of C1QBP with CD44 or RTKs [36,39]. One possible mechanism could be that the binding of M36 to p32//C1QBP could play a pivotal role in the maintenance of mitochondrial functions as a regulator of the biosynthesis of proteins encoded by the mitochondrial genome [3,17], alongside its interaction with proteins closely related to several critical mitochondrial functions such as Parkin [41,42,51]. Our results clearly show that the pharmacological inhibition of C1QBP affected not only the levels of proteins that are crucial in mitochondrial dynamics such as DRP-1, OPA-1, and Mitofusin-2, but also the levels of Parkin protein, which is crucial in the induction of mitophagy. Additionally, in our study, cells treated with M36 showed altered mitochondrial potential and a more punctuated and fragmented mitochondrial profile than the control cells. These results are consistent with those of Noh et al. (2020) [19], who reported that the knockdown of p32/C1QBP in MEFs caused the fragmentation of the mitochondria. Similarly, the depletion of C1QBP was shown to induce mitochondrial fragmentation in HeLa cells [18] and neurons [51]. We confirmed by means of TEM that p32/C1QBP inhibition with M36 produced severe mitochondrial damage, demonstrating M36′s vital role in maintaining mitochondrial homeostasis. Indeed, in the past decade, many studies have shown that mitochondrial dynamics play a crucial role in the development and progression of cancer [42,43]. Therefore, the key role of p32/C1QBP as a regulator of this important function could, at least partially, explain its oncogenic roles in colon cancer.

Emerging experimental evidence supports the cytoprotective role of p32/C1QBP under stress conditions by regulating ULK1 and Parkin stability, both key proteins in inducing mitophagy [42,43]. However, although we observed a negative effect on Parkin levels as a result of C1QBP inhibition with M36, we did not observe any significant change in the phosphorylation of ULK1 levels in Ser757 induced by mTORC1 or by p38 MAPK, which is known to repress the autophagy process [45]. Interestingly, we also did not observe any changes in the LC3 II/I ratios and p62 protein levels of macro-autophagic markers in cells treated with M36, indicating that p32/C1QBP participates only in the regulation of mitophagy, not in the general cellular process of autophagy. In addition, we recently reported that the knockdown of p32/C1QBP sensitizes colon cancer cells to cell death induced by oxidative stress and chemotherapeutic agents, but not to cell death induced by nutritional stress, which potently induces autophagy in colon cancer cells [28]. A possible explanation could be that acute nutritional stress mainly induces autophagic cell death (autosis), while p32/C1QBP regulates the death process associated with mitochondria such as apoptosis and mitophagy, but does not participate in autosis cell death. In addition, the B-RAF- and KRAS-driven colon cancer cells used in our analyses were “addicted to autophagy” because they overexpressed the hypoxia-inducible factors HIF-1α, HIF-2α, and HIF-3α, which promote autophagy. These factors play essential roles in regulating the cell death pathways to promote cell survival [53].

Additional studies will be necessary to further understand the mechanisms underlying the cytoprotective role of p32/C1QBP. Nevertheless, our data indicate that M36 treatment induces severe mitochondrial damage and blocks cell proliferation, demonstrating that p32/C1QBP is an important therapeutic target in colon cancer.

To the best of our knowledge, this study represents the first time that a small molecular inhibitor of p32/C1QBP was evaluated in colon cancer cells. As reported, the severe cytostatic effects induced by p32 inhibition sensitizes cells to oxidative stress (radiotherapy), and chemotherapeutic agents. Thus, combined with autophagy inhibitors, p32/C1QBP inhibition shows promise as an effective colon cancer treatment. Nevertheless, in vivo experiments in animal or organoid models are needed to confirm the efficacy and safety of M36.

As with many other targets that are tumor-associated but not tumor-specific, there is concern about the adverse effects that therapies such as M36 may have on non-tumor cells. In this regard, the main factor that justifies the application of p32/C1QBP as a therapeutic target is that the protein is significantly overexpressed in many types of adenocarcinomas compared with their non-malignant counterparts. Therefore, even when the protein is present at certain levels in almost all body tissues, the detected levels of this protein will be much higher in cancer tissues. Consequently, a drug targeting this protein will accumulate predominantly in the tumoral regions, which can reduce the side effects of the drug compared with other chemotherapies that attack more general cellular mechanisms or proteins. The findings made to date with M36 and with other targeted therapies against p32/C1QBP, such as monoclonal antibodies and tumor-homing peptides, have been acquired using both in vitro and in vivo models. These studies have focused not on the adverse effects of the drugs but rather on their impacts on tumor cells or tumorigenesis. Hence, future studies should further explore this area using both animal models and clinical assays.

## 4. Materials and Methods

### 4.1. Antibodies and Reagents

The catalog numbers and the dilutions used for the antibodies used in this work can be found in Appendix A. The antibodies used were as follows: rabbit anti-Akt, mouse anti-Phospho-Akt antibodies, mouse anti-DRP1, mouse anti-OPA1, and mouse anti-Parkin (PRK8) antibodies, which were purchased from Santa Cruz Biotechnology (Dallas, TX, USA). Rabbit anti-cleaved PARP, rabbit anti-cleaved caspase-3, rabbit anti-p21, rabbit anti-phospho-mTOR, rabbit anti-mTOR, mouse anti-Phospho-p70 S6 kinase, rabbit anti-p70 S6 kinase, rabbit anti-Phospho ERK1/2, rabbit ERK 1/2, rabbit anti-4E-BP1, rabbit anti-Phospho-4E-BP1, rabbit anti-Phospho-ULK1 (Ser757), and rabbit anti-PINK1 were all obtained from Cell Signaling Technology (Danvers, MA, USA). The FITC-conjugated goat anti-mouse antibody was obtained from Jackson ImmunoResearch (West Baltimore Pike, PA, USA). Goat anti-mouse and anti-rabbit IgG-horseradish peroxidase-conjugates were obtained from Pierce (Rockford, IL, USA). Mouse-anti-p32/C1QBP and mouse monoclonal anti-Mitofusin-2 antibodies were purchased from Abcam (Cambridge, MA, USA). Mouse monoclonal anti-actin was produced in the laboratory of Dr. José Manuel Hernandez-Hernández (CINVESTAV-IPN, Mexico City, Mexico). The CellTrace™ CFSE (C34554) reagent was purchased from Thermo Fisher Scientific (Waltham, MA, USA). The M36 inhibitor was purchased from MedChemExpress (MCE) (Concord, CA, USA), and the MitoTracker^®^ Red CMXRos probe was obtained from Invitrogen (Eugene, OR, USA).

### 4.2. Cell Culture

The colon cancer cell lines used in this work were as follows: the 112 CoN non-malignant colon cell line and the RKO, HCT116, SW480, and SW620 colon adenocarcinoma cell lines. 112CoN, RKO, and HCT116 cells were cultured in Dulbecco’s modified Eagle’s medium (DMEM) supplemented with 10% Fetal Bovine Serum (FBS), antibiotics (200 mg/mL Streptomycin and 120 mg/mL Penicillin), and 2 mM L-glutamine. SW480 and SW620 cells were maintained in DMEM F-12 supplemented with 10% FBS, antibiotics, and 2 mM glutamine. All cells were obtained from American Type Culture Collection (ATCC) (Manassas, VA, USA). They were authenticated in June 2018 using a Short Tandem Repeat DNA profiling analysis performed at the Instituto Nacional de Medicina Genomica (INMEGEN), Mexico City.

### 4.3. Viability Assays

MTT assays were used to evaluate viability. For these assays, 2.5 × 10^3^ RKO, HCT116, SW480, SW620, 112CoN, or RKO sh-p32 cells/well were seeded on a 96-well plate and grown for 24 h. Next, the cells were treated with an M36 inhibitor at concentrations of 0, 3.1, 6.25, 12.5, 25, 37.5, 50, 100, and 150 µM for 72 h to assess the effects on cell viability. After treatments, 0.5 mg/mL MTT was added to the cells for 3 h at 37 °C, with the mixture protected from light. MTT-formazan crystals were dissolved in acid isopropanol at pH = 4 and quantified via spectrophotometry at 570 nm.

Knockdown of p32/C1QBP in RKO cells was achieved via stable transfection of the RNAi-Ready pSIREN-RetroQ plasmid from Clontech, which generated the shRNA for the p32/C1QBP protein. As a control, RKO cells were stably transfected with the empty plasmid.

### 4.4. MitoTracker Assay

RKO cells were seeded on coverslips and treated for 72 h with DMSO as a vehicle or the M36 inhibitor. For staining, a MitoTracker^®^ Red CMXRos probe (M7512, Invitrogen) was diluted at a working concentration of 250 nM in a fresh growth medium, and was prewarmed at 37 °C. The growth medium of the cultures was carefully removed, and the growth medium supplemented with a MitoTracker probe was added. The cells were incubated for 10 min at 37 °C. After staining, the growth medium was removed, and the cells were fixed in a prewarmed solution of 1% paraformaldehyde (PFA) for 5 min at 37 °C. Subsequently, the cells were permeabilized and blocked with 1X PBS, 0.3% Triton X-100, and 10% SFB for 1 h at room temperature. Then, the cells were incubated overnight at 4 °C with mouse anti-p32 (C1QBP) antibody diluted 1:100 in 1X PBS, 0.1% BSA, and 10% SFB. Next, the cells were incubated for 2 h at room temperature with FITC-conjugated goat anti-mouse antibody diluted 1:200 in 1X PBS, 0.1% BSA, and 10% SFB. Subsequently, the nuclei were stained using a solution containing DAPI diluted in PBS 1X with 0.05% Triton X-100. After washing, the coverslips were mounted with the Vectashield anti-fade reagent. Cell fluorescence was examined using a confocal microscope (Nikon A1R+ STORM).

### 4.5. Confocal Immunofluorescence Microscopy

Confocal images were acquired using confocal microscopy with a Nikon A1R+ STORM. Analysis of the images was carried out using the software NIS Elements Viewer (https://www.nis-elements.cz/en, accessed on 1 January 2024) and Image J software (version 1.47b) obtained from the National Institutes of Health website (http://imagej.nih.gov/ij/, accessed on 1 January 2024).

### 4.6. Colony Formation Assay

To perform colony formations assays, 1 × 10^2^ RKO, HCT116, SW480, or SW620 cells per well were seeded in 96-well plates. Next, the cells were cultured at 37 °C for 7 days. After that time, the obtained colonies were fixed with absolute methanol and stained with a 0.5% violet crystal- 25% methanol solution. The number of colonies was quantified under a light microscope.

### 4.7. Proliferation Assay

Proliferation was measured by labeling the cells with Carboxyfluorescein Diacetate Succinimidyl Ester (CFSE) before pharmacological treatment to trace the generation of cells via dye dilution. Cells were incubated with CFSE 1 µM in PBS for 20 min at 37 °C, followed by washing the cells using a DMEM medium with SFB 10%. Cells were treated with or without the M36 inhibitor for six days and analyzed in an Attune NXT flow cytometer.

### 4.8. Western Blot

Cells were lysed with a RIPA lysis buffer (50 mM Tris-HCl pH = 7.4, 150 mM NaCl, 0.1% SDS, 1% NP-40, 0.25% Na-deoxycholate, 1 mM EDTA) supplemented with protease and phosphatase inhibitors for 15 min at 4 °C. After centrifugation (13,000 rpm) for 15 min at 4 °C, 40 μg of the whole-cell lysate (supernatant) was separated using 8, 10, or 15% SDS-polyacrylamide gel electrophoresis (SDS-PAGE) followed by electrophoretic transfer to nitrocellulose membranes (Bio-Rad, Hercules, CA, USA). The membranes were blocked with 3% BSA in TBS and incubated overnight at 4 °C with the corresponding primary antibody. Detection was achieved using a SuperSignal Kit (Pierce) with a horseradish-peroxidase-conjugated secondary antibody. Actin was used as a control for equal loading.

### 4.9. Evaluation of the Akt/mTOR and MAPK Cell Signaling Pathways

In total, 3 × 10^5^ RKO cells were seeded on a 6-well plate and grown for 24 h. After that time, the cells were starved for 24 h with serum-free media and were pre-treated with the vehicle or M36. The cells were then stimulated with 10% FBS-DMEM for 3 or 6 h in the presence or absence of the vehicle or M36 inhibitor to evaluate the activation of the Akt/mTOR/p70 S6 kinase and MAPK cell signaling pathways via Western Blotting. The cells were then lysed, and the levels of phosphorylated Akt, mTOR, p70-S6K, 4E-BP1, ULK1, and ERK proteins were evaluated using Western blotting. In each case, the levels of the same proteins (non-phosphorylated) were used as a control. Actin was also used as a control for equal loading.

### 4.10. Transmission Electron Microscopy (TEM)

Cells were fixed in PBS with 2.5% glutaraldehyde/paraformaldehyde for 45 min. The cells were washed and post-fixed with 1% osmium tetroxide (OsO4). Then, the cells were dehydrated with an ascending series of ethanol and embedded in Epon 812 epoxy resin (Sigma, St. Louis, MO, USA). Ultrathin (~70 nm) sections were cut with an ultramicrotome (Leica Microsystems GmbH, Wetzlar, Germany), collected on copper grids, and stained with uranyl acetate and lead citrate. Samples were observed on a JEOL 10/10 transmission electron microscope (JEOL, Tokyo, Japan).

### 4.11. Statistical Analysis

All experiments were performed at least three times using different cell preparations. The data are expressed as the mean ± standard error of the mean (SEM). Statistical data analysis was performed using Student’s *t*-test or a one-way ANOVA with Bonferroni’s multiple comparison test. All statistical analyses were performed using GraphPad Prism 6. A value of *p* < 0.05 was considered statistically significant.

## Figures and Tables

**Figure 1 ijms-25-02712-f001:**
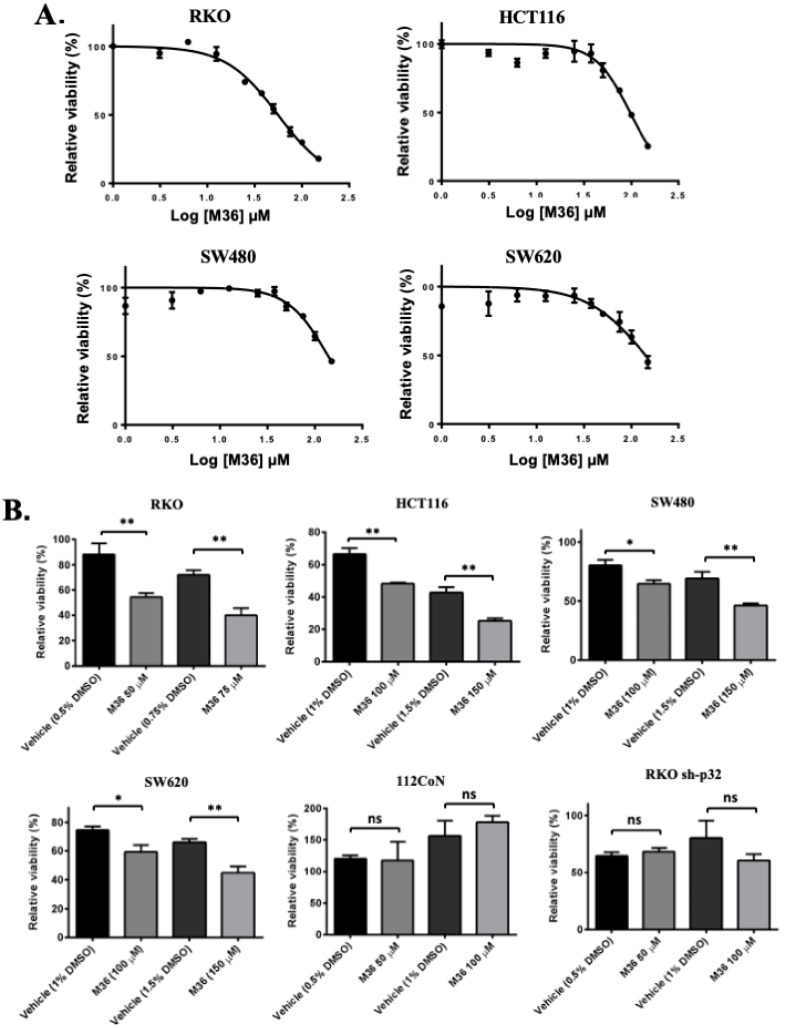
**Pharmacological inhibition of p32 protein negatively affects the viability rate of colon cancer cells.** (**A**) RKO, HCT116, SW480, or SW620 cells were seeded and cultured for 24 h. The cells were then treated with increasing concentrations of M36 inhibitor. After 72 h, the MTT assay was performed as described under Materials and Methods section. The data were fitted to a dose-response inhibition curve and the IC50 was estimated for each cell line. The estimated IC50s are IC50 (RKO) = 55.86 μM, IC50 (HCT116) = 96.95 μM, IC50 (SW480) = 138.3 μM, and IC50 (SW620) = 141.8 μM. Each point on the curve represents the mean ± SEM of three independent experiments. (**B**) A curve of viability of cells treated with the vehicle DMSO at the correspondent percent was developed for every cell line. The bar graphs show the comparison between the viability of cell lines treated with M36 versus vehicle. Non-malignant colon cells (112CoN) and p32 knockdown RKO cells were used as controls. The data are presented as the mean values + SEM from three independent experiments, * *p* < 0.05, ** *p* < 0.01; ns means non significative.

**Figure 2 ijms-25-02712-f002:**
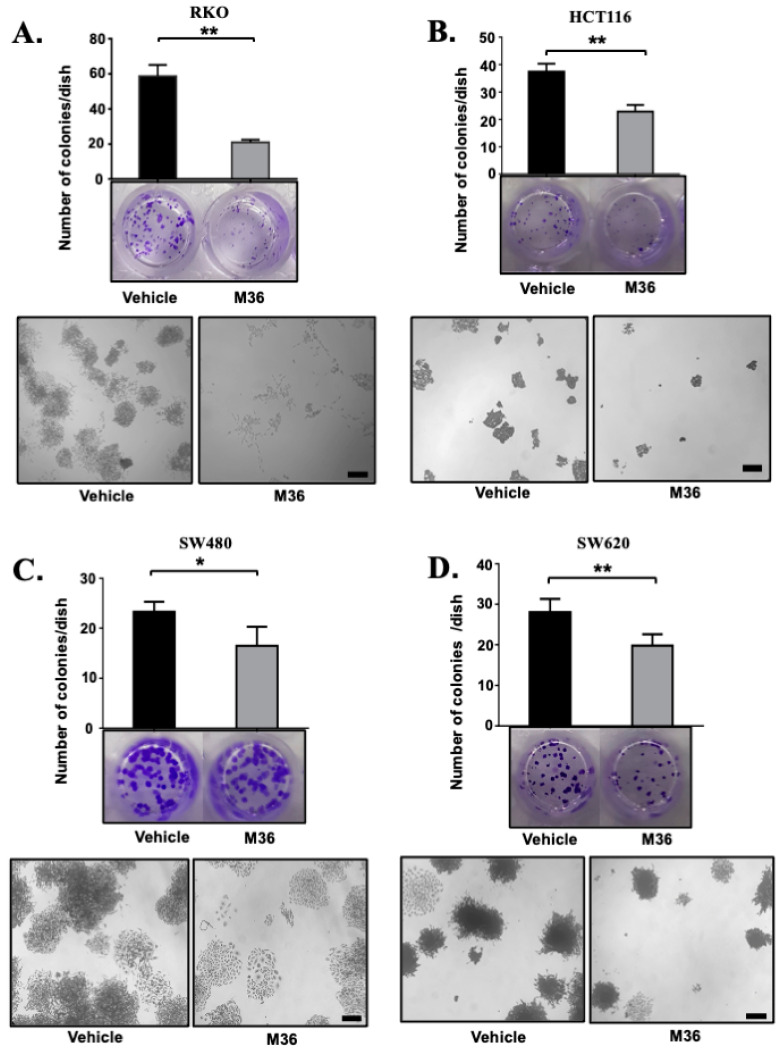
**Pharmacological inhibition of p32 negatively affects the clonogenic capacity of colon cancer cells.** Colony formation assays for RKO (**A**), HCT116 (**B**), SW480 (**C**), and SW620 (**D**) treated with vehicle or M36 inhibitor (at their respective IC50 concentrations) were performed as described in the Section 4. Representative photographs and micrographs from 3 independent experiments are shown. The data are presented as the mean values ± SEM from three independent experiments, * *p* < 0.05, ** *p* < 0.01. Scale bar 100 µM.

**Figure 3 ijms-25-02712-f003:**
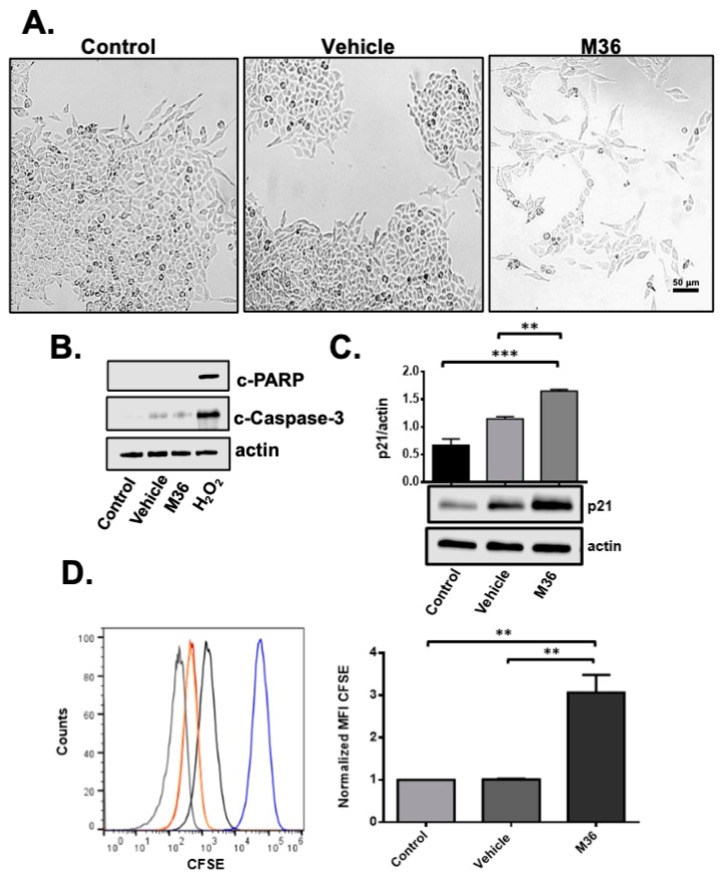
**Pharmacological inhibition of p32 induces a cytostatic but not cytotoxic effect on RKO colon cancer cell line.** (**A**) Photomicrographs of RKO colon cancer cells treated for 72 h with vehicle or M36 inhibitor (at IC50 = 56 µM), with respect to the non-treated control. The photomicrographs shown here are representative of 5 independent experiments. Scale bar, 50 µM. (**B**) The activation of PARP and Caspase 3 proteins by proteolytic cleavage in RKO cells treated with vehicle, M36, or H_2_O_2_ (positive control) with respect to the non-treated control was evaluated using Western blot. Actin was used as a control for equal loading. (**C**) The level of p21 protein in RKO cells treated with vehicle or M36 with respect to control non-treated was evaluated by Western blot. Actin was used as a control for equal loading. Densitometric analysis was performed to quantify the change in p21 expression levels. The data are presented as the mean values ± SEM from three independent experiments, ** *p* < 0.01, *** *p* < 0.001. (**D**) RKO cells were treated for 6 days with vehicle or M36 inhibitor after staining with CFSE performed as described in the Section 4. The measurement of CFSE staining intensity was performed by flow cytometry. The histogram presented is representative of three independent experiments. Blue line cells were stained with CFSE just before the measurement, gray line cells were not stained, black line were cells stained and treated with M36, orange line cells were stained and treated with vehicle, and red line cells were stained and were not treated. Mean fluorescence intensity was normalized with respect to the control. The data are presented as the mean values ± SEM from three independent experiments, ** *p* < 0.01.

**Figure 4 ijms-25-02712-f004:**
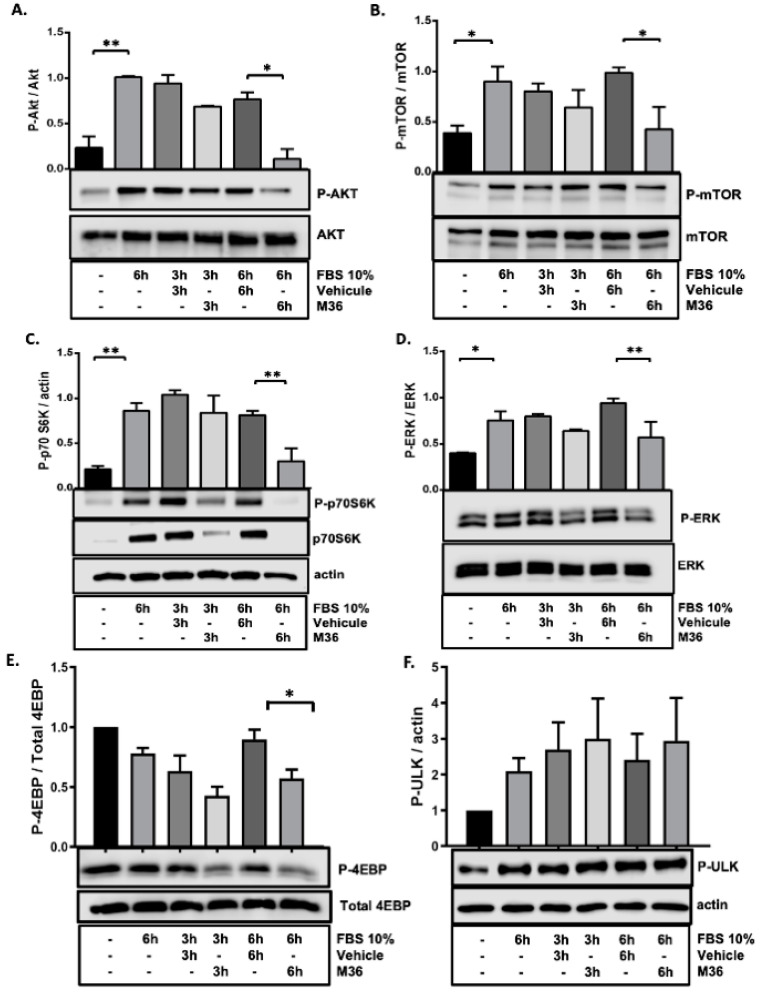
**Pharmacological inhibition of p32 negatively affects the activation of the Akt-mTor and MAP kinases signaling pathways in RKO colon cancer cells.** RKO cells were seeded on a 6-well plate and grown for 24 h. The cells were starved for 24 h with serum-free media and were pre-treated with vehicle or M36. Next, cells were stimulated with 10% FBS-DMEM for 3 or 6 h in the presence or absence of vehicle or M36 inhibitor (at IC50 = 56 µM) to evaluate the activation of Akt (**A**), mTOR (**B**), and the phosphorylation of p70-S6 kinase (**C**), ERK (**D**), 4EBP1 (**E**), and ULK1 (**F**) proteins using Western blot. Densitometric analysis was performed to quantify the levels of both phosphorylated and total proteins. The levels of each phosphorylated protein were normalized with respect to its corresponding total protein. The data are presented as the mean values ± SEM from three independent experiments, * *p* < 0.05, ** *p* < 0.01.

**Figure 5 ijms-25-02712-f005:**
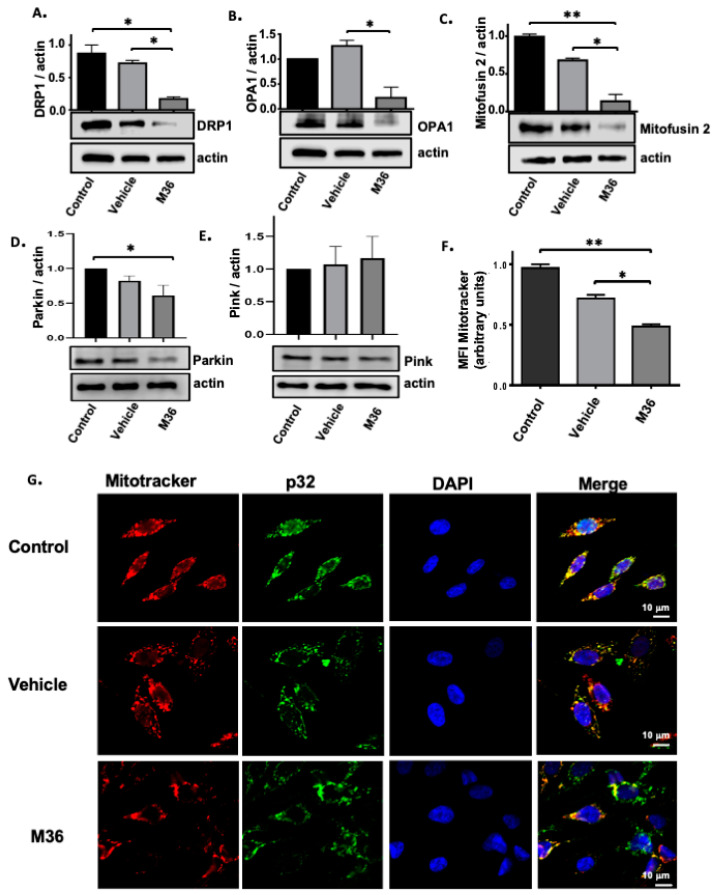
**Pharmacological inhibition of p32 negatively affects the levels of proteins involved in mitochondrial dynamics and reduces the mitochondrial mass and integrity of RKO colon cancer cells.** The levels of DRP1 (**A**), OPA1 (**B**), Mitofusin-2 (**C**), Parkin (**D**), and PINK (**E**) proteins in RKO cells treated with vehicle or M36 (at IC50 = 56 µM) with respect to non-treated control was evaluated by Western blot. Actin was used as a control for equal loading. Densitometric analysis was performed to quantify the change in protein expression levels. The data are presented as the mean values ± SEM from three independent experiments, * *p* < 0.05, ** *p* < 0.01, (**D**) RKO cells were treated for 72 h with vehicle DMSO or M36 inhibitor (at IC50 = 56 µM). (**F**,**G**). After treatment, a fresh growth medium complemented with a MitoTracker^®^ Red CMXRos probe was added to cells to label mitochondria (red). Detection of p32 was performed by indirect immunofluorescence using a FITC-conjugated antibody (green), and the nuclei were stained with DAPI (blue). The levels and cell distribution of p32, MitoTracker, and DAPI were examined by immunofluorescence confocal microscopy. Representative images from 3 independent experiments are shown in panel (**G**). The mean fluorescence intensity of MitoTracker in arbitrary units from analysis of immunofluorescence confocal microscopy images is also shown in panel (**F**). The data are presented as the mean values ± SEM from at least three independent experiments, * *p* < 0.05, ** *p* < 0.01. Scale bar, 10 μm.

**Figure 6 ijms-25-02712-f006:**
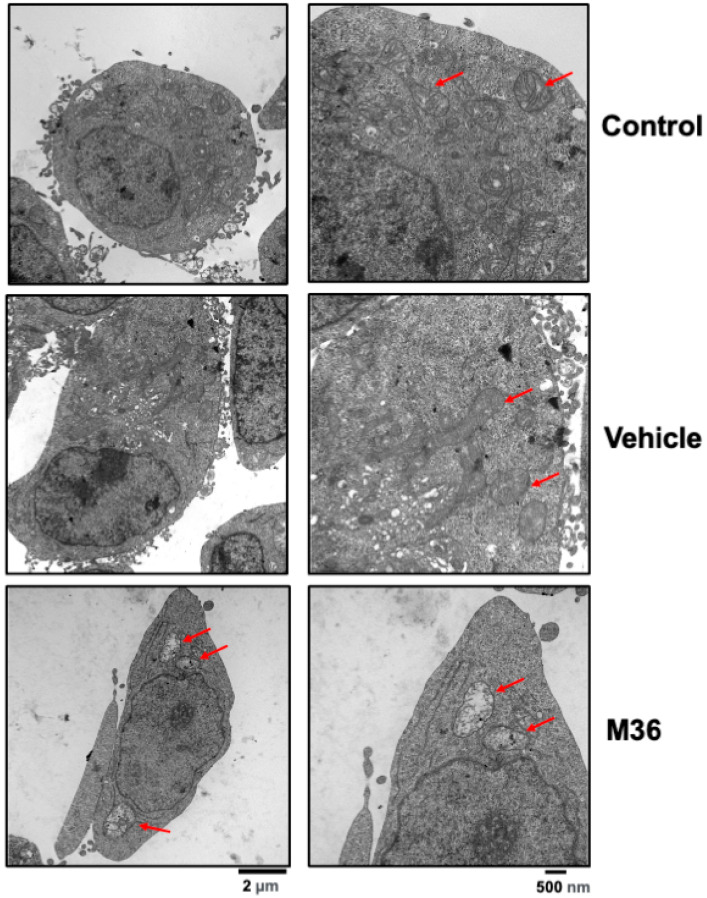
**Pharmacological inhibition of p32 induces mitochondrial damage in RKO colon cancer cells.** RKO cells were washed, fixed, and prepared and stained to visualize the mitochondrial morphology as described in the Section 4. Representative electron micrographs of RKO colon cancer cells obtained after 72 h treatment with 56 µM M36 (lower panel) compared with controls (upper and middle panels) are shown. The arrows indicate the damaged mitochondria. Scale bars, 500 nm and 2 μm.

## Data Availability

Data are contained within the article.

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
