# Peer review of "Inhibition of Multifunctional Protein p32/C1QBP Promotes Cytostatic Effects in Colon Cancer Cells by Altering Mitogenic Signaling Pathways and Promoting Mitochondrial Damage"

_ijms, 2024, doi:10.3390/ijms25052712_

Round 1

Reviewer 1 Report (New Reviewer)

Comments and Suggestions for Authors

Here, the authors report the effects of a small molecule inhibitor of p32, a protein that is overexpressed in colon cancer and promotes malignancy traits. The authors show that the inhibitor M36 reduces the viability, proliferation, and clonogenicity of colon cancer cells by impairing the activation of Akt-mTOR and MAPK pathways and inducing mitochondrial dysfunction. The article suggests that p32 is an important oncogenic target and that M36 is a potential therapeutic drug for colon cancer.

Main points: 

The article provides evidence for the role of p32 in colon cancer and the mechanisms of action of M36.

The article uses multiple methods and cell lines to validate the results, showing the strength of the conclusions. 

Minor points: 

The article does not include in vivo experiments to confirm the efficacy and safety of M36 in animal models or organoid models. Could the authors discuss what could be the toxicity of such a drug for normal cells, compared to cancer cells, as C1QBQ is expressed in healthy tissue https://www.proteinatlas.org/ENSG00000108561-C1QBP/tissue? 

Could they discuss whether M36 coukd be tested in other cancer types?

Could they also compare M36 with other existing or potential drugs for colon cancer? What would be the advantage of this drug compared to the currently approved ones? Any suggestions for combined therapy.

C1QBP dependency appears to be correlated to some drug resistance patterns in Col. adenocarcinoma cells: https://depmap.org/portal/interactive/?filter=slice%2Fcontext%2FColon%2520Adenocarcinoma%2Flabel&regressionLine=true&associationTable=false&x=slice%2FChronos_Combined%2F2702%2Fentity_id&y=slice%2FRep_all_single_pt%2F48740%2Fentity_id&color=

Please use C1QBP instead of p32 in the text, as it is the approved alias of the gene, see https://www.genenames.org/data/gene-symbol-report/#!/hgnc_id/1243

Comments on the Quality of English Language

minor typos

Author Response

Minor Points

  1. The article does not include in vivo experiments to confirm the efficacy and safety of M36 in animal models or organoid models. Could the authors discuss what could be the toxicity of such a drug for normal cells, compared to cancer cells, as C1QBQ is expressed in healthy tissue https://www.proteinatlas.org/ENSG00000108561-C1QBP/tissue?

M36 is a specific inhibitor of the p32/C1QBP protein, as has been demonstrated by Yenugonda et al. in 2017 for glioma cells and in the present work for colon cancer cells. As a specific inhibitor, the effects of the drug are more significant for cells overexpressing the protein, as is the case of glioma and colon cancer cells. In both works, it has been demonstrated that the effects exerted by M36 on the viability of cells are significantly lower in cells with a low expression of C1QBP (normal tissue or non-malignant cells). For example, in our work, the effects exerted by M36 on the viability of the 112CoN cell line (non-malignant colon cells with a relatively low expression of C1QBP) were non-significative compared to the vehicle-treated group.

The main reason that justifies the usage of p32/C1QBP as a therapeutic target is that the protein is significantly overexpressed in many types of adenocarcinomas with respect to their non-malignant counterparts. Therefore, even when the protein is present at certain levels in almost all body tissues, the levels of its detection will be much higher in cancer tissues. Consequently, a drug targeted against this protein will accumulate mainly in the tumoral regions, which can reduce the side effects of the drug concerning other chemotherapies that attack more general cellular mechanisms or proteins.   However, as occurs with many other targets that are tumor-associated but not tumor-specific, there is concern about the adverse effects that therapies such as M36 may have on non-tumor cells. Therefore, toxicity and adverse effects tests must be carried out during the preclinical phase and the clinical studies. The findings made to date with M36 and other targeted therapies against p32/C1QBP as monoclonal antibodies or tumor-homing peptides have been carried out both in vitro and in vivo models. They have not focused on the adverse effects of the drugs but rather on their impact on tumor cells or tumorigenesis. Hence, future studies should be carried out in this direction in animal models and clinical assays.

We have explained these issues in the discussion in the revised version of the manuscript.

  1. Could they discuss whether M36 could be tested in other cancer types?

M36 has been tested only in two cancer types so far: glioma and colon cancer. However, considering that the protein is overexpressed and promotes malignancy traits in several adenocarcinoma types such as breast, lung, prostate, pancreas, and hepatic cancer (Reviewed in: Egusquiza-Alvarez A and Robles-Flores M. J Cancer Res. Clin. Oncol. 148, 1831-54, 2022), there are high probabilities of success using M36 against malignancy traits in the aforementioned cancer types.

  1. Could they also compare M36 with other existing or potential drugs for colon cancer? What would be the advantage of this drug compared to the currently approved ones? Any suggestions for combined therapy?

C1QBP dependency appears to be correlated to some drug resistance patterns in Col. adenocarcinoma cells: https://depmap.org/portal/interactive/?filter=slice%2Fcontext%2FColon%2520Adenocarcinoma%2Flabel&regressionLine=true&associationTable=false&x=slice%2FChronos_Combined%2F2702%2Fentity_id&y=slice%2FRep_all_single_pt%2F48740%2Fentity_id&color=

As mentioned before, the main advantage of this drug with respect to other approved ones is that it is targeted against a protein that is overexpressed in colon cancer cells and, at the same time, is promoting chemoresistance, tumorigenesis, clonogenicity, and other malignancy traits of colon cancer cells. Previously, we have shown that p32/C1QBP is a promoter of chemoresistance of colon cancer cells to 5FU and CCI-779 (mTor inhibitor Temsirolimus). Those results, together with the evidence presented in the present work, strongly suggest that the combination of M36 with CCI-779 or 5FU is promising for treating colon adenocarcinoma. Probably the combination of M36 and CCI-779 or 5FU will reduce the resistance rates of certain colon cancer subtypes to the aforementioned established chemotherapies.

  1. Please use C1QBP instead of p32 in the text, as it is the approved alias of the gene; see https://www.genenames.org/data/gene-symbol-report/#!/hgnc_id/1243.

Done. We changed p32 alone for p32/C1QBP  or C1QBP all over the text, including the title.

Reviewer 2 Report (New Reviewer)

Comments and Suggestions for Authors

The authors demonstrate the ability of the p32 inhibitor, M36, significantly decreases the viability rate, clonogenic capacity, and proliferation rate of different colon cancer cell lines. The effects of the M36 is cytostatic and induces a decreased activation of Akt-mTOR and MAPK in RKO colon cancer cells. Additionally, the treatment with M36 significantly affects the malignant cells' mitochondrial integrity and dynamics, indicating that p32 plays an essential role in maintaining mitochondrial homeostasis.

In general, the paper needs to be improved.

1) The authors used M36 as inhibitor of p32, but they did not demonstrate  the effective inhibition of p32.

2) The authors stated that RKO cells have the highest level of p32 without showing any data or reference that demostrate it.

3) The authors used EC 50 and IC 50 wrongly as synonyms. In this study it is appropriate to calculate the IC 50 of M36.

4) In the abstract the authors stated  that M36 is a potential therapeutic drug for the treatment of colon cancer.. Nevertheless, the evidence obtained in this study are not so strong to suggest a potential use in clinic trials. Please correct accordingly the sentence also in the introduction.

5) Please , in the "methods" section, indicate in a table the dilution of primary and secondary antibodies used in the western blots.

6) Fig 1A:  please,  remove the parentheses in the x axis of the graph

7) Fig 1 and fig. 4: please move the description of methods  or the results ( e.g. IC50) described in the caption, in the corresponding sections.

8) Fig. 2 : please, correct the y axis title in the graph with "number of colonies/dish".

Fig. 3A: In the caption, it should be used photomicrograph instead of micrograph

8) Fig. 5G: the quality of  images presented is too low to assess the fragmentation rate of mithocondria. 

9) Fig. 5 : please, align the caption with the figure.

10) In the "discussion" section, please remove references to figures.

11) The "references" section need to be carefully checked, in order to use the same criteria to write all references.

Comments on the Quality of English Language

Moderate editing of English language is required

Author Response

1.The authors used M36 as inhibitor of p32, but they did not demonstrate the effective inhibition of p32.

M36 is commercialized as a specific inhibitor of p32 by MedChemExpress (MCE). For more information, please refer to the Product Data Sheet Cat. No. HY-124718.  The authors that developed the inhibitor (Yenugonda et al., 2017, reference #43) showed evidence of the inhibitor's capacity and specificity of M36 against p32. For instance, M36 is able to inhibit the binding of p32 to the globular head of C1q and the home-tumor peptide Lyp. Although it is not easy to demonstrate the specific inhibition of p32 due to its multifunctionality and the high amount of ligands it has, especially in cancer cells, in this work, we show that M36 can mimic several effects caused by the knockdown of p32 in colon cancer cells such as: decreasing the clonogenic capacity and the rate of activation of pathways downstream of RTKs in RKO cells. 

  1. The authors stated that RKO cells have the highest level of p32 without showing any data or reference that demonstrate it.

We apologize for this omission. We have provided this reference in the 3.1 Results section when we stated this. We have previously reported the expression levels of p32/C1QBP in several colon cell lines both by Western blotting and by immunofluorescence assays (Reference #34: Egusquiza et al., Front Oncol. 2021, 11:1–15). 

  1. The authors used EC 50 and IC 50 wrongly as synonyms. In this study, it is appropriate to calculate the IC 50 of M36.

We apologize for the error. We used IC50 in all calculations.

  1. In the abstract, the authors stated that M36 is a potential therapeutic drug for the treatment of colon cancer. Nevertheless, the evidence obtained in this study is not so strong to suggest a potential use in clinical trials. Please correct accordingly the sentence also in the introduction.

We agree with you. We have corrected the last sentences of the Abstract and the Introduction to say that M36 may be a promising therapeutic drug for the treatment of colon cancer. In addition, we have mentioned now in the discussion section that in vivo experiments will be needed to confirm the efficacy and safety of M36 in animal models and in clinical assays.

  1. Please, in the "methods" section, indicate in a table the dilution of primary and secondary antibodies used in the western blots.

All the information about the antibodies used in our studies, including the dilutions used and the catalog numbers, are now presented in Supplementary Table I. 

  1. Fig 1A:  please remove the parentheses in the x axis of the graph

Done

  1. Fig 1 and fig. 4: please move the description of methods or the results ( e.g. IC50) described in the caption, in the corresponding sections.

We removed the description of methods but not the IC50, attending to the suggestion made by other reviewers.

  1. Fig. 2 : please correct the y-axis title in the graph with "number of colonies/dish".

The y axis was corrected following your suggestion.

  1. Fig. 3A: In the caption, it should be used photomicrograph instead of micrograph

The micrograph was changed by photomicrograph in the caption of Figure 3A.

  1. Fig. 5G: the quality of images presented is too low to assess the fragmentation rate of mitochondria.

You are right. Although we have provided better-quality images, they only allow us to observe signs of mitochondrial fragmentation. That was the reason we then analyzed the mitochondrial morphology and integrity by TEM in Figure 6.

  1. Fig. 5 : please, align the caption with the figure.

Done.

  1. In the "discussion" section, please remove references to figures.

Done.

  1. The "references" section need to be carefully checked, in order to use the same criteria to write all references.

The “references” section was carefully checked and corrected.

Reviewer 3 Report (New Reviewer)

Comments and Suggestions for Authors

Alejandro et al. reported the cytostatic effect of the M36 inhibitor, limiting the p32 activity, in an in vitro colon cancer model. The authors showed that M36 effects are mediated via the MAPK/ERK signaling pathway. Also, M36-induced mitochondria damage has been reported.

The manuscript is generally well-designed, however, there are some issues that I need to address:

  1. 0,5% or 1,5% DMSO exerted significant cytotoxic effects itself, reducing cell viability up to 50% (HCT116 cells). Such control is not the best environment for proper results interpretation, because intracellular processes caused by toxic DMSO could significantly influence the tested inhibitor action. The authors should prepare lower stock solutions, allowing them to achieve in the experimental medium less toxic DMSO concentrations (0,05 or 0,01%). In such a case I have serious doubts about the presented data.
  2. Why did the authors treat cells with various M36 concentrations, instead unified experimental design, 50 uM, 75 uM, 100 uM, and 150 uM for all tested cell lines. Currently, it is misleading when you want to compare the graphs.
  3. Did you verify the statistical differences between low, and high-concentration groups? It seems that in 112CoN cells, M36 dose-dependently stimulates cell viability. The authors did not discuss this observation. 
  4. In all WB figures, the tested P position should be clearly marked. 
  5. Figure 6- why the cells were treated with 56 uM M36? Is it IC50 value? Please, provide M36 IC50 values for all tested cell lines below Figure 1. 
  6. Please provide the information about AB dilutions for WB analysis along with cat. no. in the Materials and Methods section. 

Minor issue:

  1. The abbreviations should be explained whenever they appear first time in the text, please update (p.1, 2)
  2. The References style does not follow the Journal template.
  3. The citation in the text should be in [] not ().
  4. Please fill in the paragraph: Data availability statement.
  5. Extensive English editing is required.

Comments on the Quality of English Language

Extensive editing of English language required

Author Response

0.5% or 1.5% DMSO exerted significant cytotoxic effects itself, reducing cell viability up to
50% (HCT116 cells). Such control is not the best environment for proper results interpretation, because intracellular processes caused by toxic DMSO could significantly influence the tested inhibitor action. The authors should prepare lower stock solutions, allowing them to achieve in the experimental medium less toxic DMSO concentrations (0,05 or 0,01%). In such a case, I have serious doubts about the data presented.

All the experiments performed in our work were developed using non-treated control cells and cells treated with the same DMSO percentage concentration without M36 as a reference. In addition, all the results presented are statistically significant for cells treated with M36 compared to the cells treated with the respective concentration of DMSO. Even when DMSO exerts certain effects on the viability of colon cancer cells like HCT116, the effects caused by M36 are statistically and biologically significant for all the cell lines tested and in the different cellular functions assayed. In particular, for RKO, the cell model used for most of the experiments performed here, the effects exerted by DMSO on the functions that were evaluated at the concentration used (0.56% DMSO) are non-significant when compared to non-treated cells.

Unfortunately, due to the low solubility of the compound in water, it is not possible to use a lower DMSO concentration. Importantly, the dose-response experiments performed here were based on the previous work published by Yenugonda et al. in 2017 (the authors that developed the inhibitor) and the datasheet of the product commercialized by MedChemExpress (MCE) Cat No. HY-124718. For more information about the product, refer to the Published Data Sheet using the Cat No. in www.MedChemExpress.com.

  1. Why did the authors treat cells with various M36 concentrations, instead unified experimental design, 50 uM, 75 uM, 100 uM, and 150 uM for all tested cell lines. Currently, it is misleading when you want to compare the graphs.

As can be verified in Figure 1 panel A, we developed M36 dose-response curves for the four cell lines evaluated in concentrations of 150, 100, 75, 50, 37.5, 25, 12.5, 6.25, 3, 0 µM (It is clearly described in the Figure 1A caption). Obviously, each cell line has a different response, considering that they have different total p32/C1QBP levels, intracellular locations, and a diverse molecular and genetic background, as well since these cell lines come from different biological sources. That is the reason why they have different IC50s when treated with M36. In panel B of Figure 1, we only show the comparison with DMSO at the same concentration for the most important points of each curve, always including the value that is closer to the IC50 of the curves presented in Figure 1 Panel A. 

  1. Did you verify the statistical differences between low and high-concentration groups? It seems that in 112CoN cells, M36 dose-dependently stimulates cell viability. The authors did not discuss this observation.

The dose responses for the different cell lines were performed as independent experiments. Therefore, a statistical comparison between cell lines is not possible. However, it is possible to compare the IC50 values between them. The lower IC50 value obtained was for RKO cells and the higher for SW480 and SW620 cells, results that, interestingly, negatively correlated with the p32/C1QBP expression levels found in every cell line (as we reported previously in Frontiers in Oncol. 11: 642940, 2021). Another way to compare the effects in the different cell lines tested is by comparing the level of significance of the difference between groups treated with M36 with respect to the group treated with DMSO. While there are no differences between groups for 112CoN cells (low expression of C1QBP) and RKO C1QBP knockdown, there is a significant effect for RKO cells and a lower significance for the cell lines with lower levels of C1QBP (SW480 and SW620).

For 112CoN, there are no significant differences between the groups treated with M36 and the groups treated with DMSO. Therefore, the effects observed for that cell line are not attributed to the inhibitor that is under study in this work but to the vehicle DMSO. It has been demonstrated that DMSO can induce the proliferation of different cell lines at certain concentrations. However, this work focuses on p32/C1QBP and its inhibitor M36. Since the effects observed on 112CoN are not related to M36, they are beyond the scope of this work. Interestingly, the lack of response of 112CoN demonstrates that the cytostatic effects of M36 are specific for cells overexpressing p32. 

  1. In all WB figures, the tested P position should be clearly marked. 

Done.

5.Figure 6- why the cells were treated with 56 uM M36? Is it IC50 value? Please, provide M36 IC50 values for all tested cell lines below Figure 1.

Correct, 56uM is the IC50 value for RKO cells treated with M36, as it is stated in the corresponding Figure 1A caption. The IC50 for the cell lines evaluated in this work is also shown in Figure 1, Panel A caption:  “The data was fit to a dose-response inhibition curve, and the IC50 was estimated for each cell line. The estimated IC50s are IC50 (RKO) = 55.86 µM, IC50 (HCT116) = 96.95 µM, IC50 (SW480) =138.3 µM, IC50 (SW620) =141.8 µM. Each point on the curve represents the mean + SEM of three independent experiments.”

  1. Please provide the information about AB dilutions for WB analysis along with cat. no. in the Materials and Methods section. 

All the information about the antibody dilutions used and the catalog numbers are now presented in Supplementary Table I. 

Minor issue

  1. The abbreviations should be explained whenever they appear first time in the text, please update (p.1, 2)

Done.

  1. The References style does not follow the Journal template.

The reference style was changed following the Journal template.

  1. The citation in the text should be in [] not ().

All citation formats used throughout the manuscript were changed accordingly.

  1. Please fill in the paragraph: Data availability statement.

Done

  1. Extensive English editing is required.

The manuscript was edited now by MDPI Services.

Round 2

Reviewer 2 Report (New Reviewer)

Comments and Suggestions for Authors

The authors have sufficiently improved the manuscript in accordance with the reviewers' suggestions.

Reviewer 3 Report (New Reviewer)

Comments and Suggestions for Authors

The authors explained all issues and updated the manuscript. I recommend its acceptance.

This manuscript is a resubmission of an earlier submission. The following is a list of the peer review reports and author responses from that submission.

Round 1

Reviewer 1 Report

Comments and Suggestions for Authors

o Line 111, there is a typo "e". Please revise accordingly and check through the manuscript. 

o Figure 4, the author did not use the internal standard to normalize and quantify the bands. What are the pros and cons of the analytical method used by the authors in the manuscript versus using the internal standard?

o The result section and data interpretation part are straight-forward but need more description. It would be better if authors could describe in one or two lines why a particular analysis is done and what they are looking for. Although, they have mentioned the purpose of some studies but not for all the parts.

o The limitation of this study should be pointed out in the discussion. 

Comments on the Quality of English Language

The language is fine. Some typo needs to be corrected. 

Author Response

Reviewer #1: Comments and suggestions for authors:

  1. In line 111, there is a typo "e". Please revise accordingly and check through the manuscript.

Done. All manuscript was checked and edited appropriately.

  1. The result section and data interpretation part are straightforward but need more description. It would be better if authors could describe in one or two lines why a particular analysis is done and what they are looking for. Although, they have mentioned the purpose of some studies but nor for all the parts.

We have followed your suggestion, and in the revised version, we have described appropriately all the experiments performed and explained better what we were looking for in each one.

  1. The limitation of this study should be pointed out in the discussion.

We have rewritten the discussion section considering we obtained new important results. We also tried to emphasize the strengths and limitations of our study.

  1. The language is fine, some typos need to be corrected.

Done.

Reviewer 2 Report

Comments and Suggestions for Authors

This manuscript “Inhibition of Multifunctional Protein P32 Promotes Cytostatic Effects in Colon Cancer Cells by Altering Mitochondrial Dynamics and Mitogenic Signaling Pathways” investigates the role of protein p32 is a multifunctional and multicompartmental homotrimer that is over-expressed in many cancer types, including colon cancer. Authors in the study reinforce that p32 is an important oncogene target and that M36 is a potential therapeutic drug for the treatment of colon cancer. The manuscript attempts at addressing an interesting question but lacks robust data for the hypothesis proposed.

There are following major concerns with manuscript:
1.Authors need to check levels of p-4ebp1, p-ULK1 to claim if mTOR pathway is active/inactive upon M36 treatment.

2. Authors need to check p-ser65-ub, p-dnm1l, p-ser65-prkn levels upon M36 treatment to claim if mitochondrial dynamics is affected inducing mitogenic signaling pathways in given treatment conditions.

3. Authors can merge figure 1 and 2 without loosing flow of manuscript for better presentation of manuscript.

4. Authors need to check levels of PINK1 and parkin accumulation in mitochondrial extract of M36 treated cells to claim if mitochondrial dynamics is altered.

5. Authors need to perform TMRE staining which can reveal if any changes in mitochondrial membrane potential are happening in response to M36 treatment.

8. Figure 1a need the control/DMSO treated cells only group.

9. Manuscript lacks in vivo study for the hypothesis proposed.
10. The authors should provide a scale bar for all respective TEM/confocal/photomicrographs.
11. The manuscript has many grammatical errors from the very beginning and should be rewritten for better understanding of manuscript.

Comments on the Quality of English Language

see comment 11 above.

Author Response

Reviewer #2: The manuscript attempts at addressing an interesting question but lacks robust data for the hypothesis proposed. There are the following major concerns with the manuscript:

  1. Authors need to check levels of p-4ebp1, p-ULK1 to claim if mTOR pathway is active/inactive upon M36 treatment.

Done. The results are presented in Figures 4B, 4C, 4E, and 4F.

  1. Authors need to check p-ser65-ub, p-dnm1, and p-ser65-prn levels upon M36 treatment to claim if mitochondrial dynamics is affected, inducing mitogenic signaling pathways in given treatment conditions.

Done. We examined the effects of p32 inhibition with M36 in the level of expression of three key proteins in the regulation of mitochondrial dynamics: DRP1, OPA1, and Mitofusin -2. We found a detectable loss of DRP1, OPA1, and Mfn2 levels upon M36 treatment of cells, indicating that the mitochondrial dynamic was affected.

  1. Authors can merge figure 1 and 2 without loosing flow of manuscript for better presentation of manuscript.

We tried to follow your suggestion, but there are too many panels to put in only one figure that must fit on one page, which would sacrifice the size and visibility of the fonts of each panel. Thus, we decided to keep separating the figures but avoid the loose flow of the manuscript describing the results appropriately.

  1. Authors need to check levels of PINK1 and parkin accumulation in the mitochondrial extract of M36 treated cells to claim if mitochondrial dynamics is altered.

To explore the effect of p32 inhibition on mitophagy, we examined the effects on the PINK-Parkin pathway, which plays an important role in regulating mitophagy. It has been demonstrated that p32 is involved in mitochondrial autophagy to remove the dysfunctional mitochondria by regulating Parkin (reference 56). In agreement with this, we show in Figure 5 that although PINK total levels were not affected by p32 inhibition, Parkin levels were decreased (Figure 5D), indicating that inhibiting p32 would impede the damaged mitochondria from being removed.

  1. Authors need to perform TMRE staining, which can reveal if any changes in mitochondrial membrane potential are happening in response to M36 treatment.

Mitochondrial membrane potential and morphology are considered key information parameters of mitochondrial functional state. They can be studied using fluorescent dyes ("probes") like TMRM and Mitotracker dyes. Using the Mitotracker Red CMXRos probe as an indicator of mitochondrial membrane potential alteration by confocal microscopy,  we found that cells treated with M36 had a decrease in the Median Fluorescence Intensity (MFI) of Mitotracker per cell compared to the cells treated with vehicle or non-treated (Figure 5F). We also observed a higher fragmentation rate in the cells treated with M36 (Figure 5G) compared to control ones, suggesting that M36 caused mitochondrial damage, which was confirmed by observing the morphology with a Transmission Electron Microscope (TEM).

  1. Figure 1a needs the control/DMSO treated cells only group

The control/DMSO vehicle-only group, without M36, corresponds in each graph of Figure 1A to the 0.0 Log {M36} mM with 100% Relative viability.

  1. Manuscript lacks in vivo study for the hypothesis proposed.

We have recently proved in vivo that knockdown of p32 affects the tumorigenic ability of colon cancer cells using a xenograft model in nude mice (Frontiers in Oncology 2021 11:642940). In this new study, we aimed to deepen into the molecular mechanisms involved in p32-mediated actions using, for the first time, a small molecular specific inhibitor (M36) to evaluate its potential use as a drug for colon cancer treatment. M36 drug has not been widely studied (as far as we know, the only research about the drug done before our study was performed by Yenugonda et al. in 2017). There are no precedents for in vivo assays with this drug. An in vivo study would take a careful series of dose-response studies, including exploring the pharmacodynamics of the drug. Therefore, an in vivo assay is outside the scope of our work and could form part of subsequent work in our laboratory in the future.

  1. The authors should provide a scale bar for all respective TEM/confocal photomicrographs.

The scale bars for all TEM/confocal/photomicrographs are now provided.

  1. The manuscript has many grammatical errors from the very beginning and should be rewritten for a better understanding of manuscript.

The manuscript was rewritten in many parts and totally edited for better understanding.